# Multicomponent molecular memory

Christopher E. Arcadia[1], Eamonn Kennedy[1], Joseph Geiser[1], Amanda Dombroski[1], Kady Oakley[1],
Shui-Ling Chen [1], Leonard Sprague [1], Mustafa Ozmen[1], Jason Sello[1], Peter M. Weber [1], Sherief Reda[1],
Christopher Rose[1], Eunsuk Kim[1], Brenda M. Rubenstein [1] & Jacob K. Rosenstein [1]*

Multicomponent reactions enable the synthesis of large molecular libraries from relatively few inputs. This scalability has led to the broad adoption of these reactions by the pharmaceutical industry. Here, we employ the four-component Ugi reaction to demonstrate that multicomponent reactions can provide a basis for large-scale molecular data storage. Using this combinatorial chemistry we encode more than 1.8 million bits of art historical images, including a Cubist drawing by Picasso. Digital data is written using robotically synthesized libraries of Ugi products, and the files are read back using mass spectrometry. We combine sparse mixture mapping with supervised learning to achieve bit error rates as low as 0.11% for single reads, without library purification. In addition to improved scaling of non-biological molecular data storage, these demonstrations offer an information-centric perspective on the high-throughput synthesis and screening of small-molecule libraries.

[1] Brown University, Providence, RI, USA. *email: jacob_rosenstein@brown.edu

Significant advances toward useful molecular-scale data systems have been made by exploiting DNA[1–8] and other sequence-defined polymers[9–11]. However, linearly ordered macromolecules represent only a tiny fraction of the near-limitless variety of chemistries, which could be used to represent information. To find alternative examples of molecular information systems, one need look no further than the simplest single-celled organisms, which have evolved to make use of many complementary forms of chemical information, including loosely ordered mixtures of small molecules, such as metabolites and dissolved ions. Similarly, recent demonstrations have stored digital data not only in DNA[4,5,12], but also using short peptides[13] and metabolites[14].

While macromolecules will continue to be important for information systems, complementary small-molecule approaches can offer a number of potential advantages[13,15]. They do not require polymerization or enzymatic steps; they can be designed to resist cellular digestion[16] and extreme environmental conditions; and they can be economical to produce. However, previous demonstrations of non-polymeric molecular data have faced capacity scaling challenges, which limited their scope to small files, such as encryption keys[17,18].

In this work, we encode millions of bits of data, in the form of digital images, using mixtures of small molecules. Rather than representing information in linear molecular sequences, we store data in locally disordered mixtures of small molecules, which can be identified by their molecular structures. This approach may appear comparatively difficult to scale to large amounts of data since we cannot simply add more subunits, as in the case of a polymer. We overcome this hurdle by creating large libraries of unique compounds through automated multi-component reactions.

We introduce a process to perform the nanoliter-scale synthesis and validation of thousands of unique Ugi products per day, without requiring purification or the use of solid supports. Some of these compounds are likely novel and have not been experimentally characterized before. To use these Ugi libraries, we have developed tools that can identify information-bearing molecules in complex chemical mixtures. By combining high-resolution mass spectrometry with supervised learning, we show how to use isotopes, adducts, impurities, and chemical interactions to improve the identification of information-carrying compounds. Additionally, we improve on previous demonstrations of non-genomic data storage by implementing a sparse data encoding scheme which dramatically reduces error rates. Furthermore, the techniques used here can be applied to other scalable chemical libraries.

An overview of the data storage process is provided in Fig. 1, which depicts the storage of a 0.88 megapixel digital image derived from a Cubist charcoal drawing of a violin by Pablo Picasso[19]. Other datasets are shown in Fig. 6. These images represent the largest amount of digital data stored in a non-polymeric molecular form (Supplementary Fig. 9).

Even in these early demonstrations, encoding between 16 and 575 bits of data per position compares favorably to some aspects of conventional memory devices, in which information is typically encoded using a single scalar parameter (e.g. charge) per location, and where electronic noise sources make it impractical to store more than a few bits per cell[20,21]. In order to further improve density, semiconductor memory is increasingly structured in three dimensions[22]. While the physical dimensions of our chemical memory spots are currently much larger than electronic memory, the concept of storing information in diverse small-molecule mixtures is valid down to the nanoscale.

In addition to the potential for dense data storage, working with large numbers of complex chemical mixtures provides opportunities to learn from information-rich annotated experimental datasets. Just as DNA memory has inspired improvements in synthesis and sequence alignment[23,24], advances in non-polymeric molecular data systems can lead to insights that may prove useful for navigating broad small-molecule spaces for drug discovery, metabolomics, and synthetic biology.

## Results

**Combinatorial library synthesis.** The automated generation of diverse non-polymeric chemical libraries is challenging because of the wide variety and complexity of synthetic protocols. To create scalable small-molecule libraries appropriate for information storage, we use the multicomponent Ugi reaction[25], which combines four reagents: an amine, an aldehyde or a ketone, a carboxylic acid, and an isocyanide, into a single product plus water. The number of unique Ugi products that can be formed scales with the number of available reagents (Fig. 2a). For instance, with 10 variations of each reagent, up to 10,000 unique multicomponent products are possible. The Ugi reaction is particularly attractive as a one-pot, single-step, room-temperature reaction[26] that has known catalysts[27,28], solid supports[29–31], accelerated conditions[32,33], and multi-step extensions[34]. Ugi reactions have been previously used as secret molecular encryption keys[17], and to create sequence-defined macromolecules[35]. Here, however, our goal is to encode millions of bits of information in mixtures of small molecules, requiring that we find efficient strategies to synthesize as many unique products as possible.

To this end, we have developed automated protocols for the high throughput synthesis of 1500 Ugi products at a time (Fig. 2b). We begin with a well plate containing five amines, five aldehydes, 12 carboxylic acids, and five isocyanides, and use an acoustic fluid handler (Echo 550, Labcyte) to enumerate all 1500 possible combinations of the four components into a 1536-well plate. After reacting, the wells are diluted to a final volume of 4 μL. Since the minimum transfer volume of the fluid handler is 2.5 nL, each library well can be dispensed more than a thousand times before it is depleted.

We initially assumed that it would be necessary to purify each library component, perhaps using solid supports, but were pleasantly surprised to find that with appropriate analysis strategies our molecular datasets could be accurately read using raw reaction solutions. Forgoing purification allowed us to streamline the experimental protocols and minimize labor and material cost, such that over the course of this work we were able to synthesize more than 10,000 compounds.

To validate the library, 20 nL from each well was analyzed with mass spectrometry (SolariX 7T, Bruker)[36,37], in matrix-assisted laser desorption ionization (MALDI) mode[38]. The Ugi product monoisotopic masses (M) are mostly between 500 and 700 Da, but we frequently observe sodiated (M + Na) and potassiated (M + K) adducts (Fig. 3b). We analyzed the library spectra for sodiated product peaks, and found that more than 90% (1346/1500) of the products had significant signal (SNR > 31.44). Additional details about library synthesis and validation are provided in the Methods and Supplementary Figs. 1–4.

**Writing data as chemical mixtures.** The composition of a chemical sample can represent abstract information, whether the sample consists of a single compound selected from a defined chemical space[17], a pool of sequence-controlled polymers[4,13], or a mixture of unique compounds[15]. With small molecule libraries, the most direct way to encode information is to use the presence or absence of each library element in a sample to represent one bit of data[14,39]. Thus, our 1500 compound Ugi libraries could encode

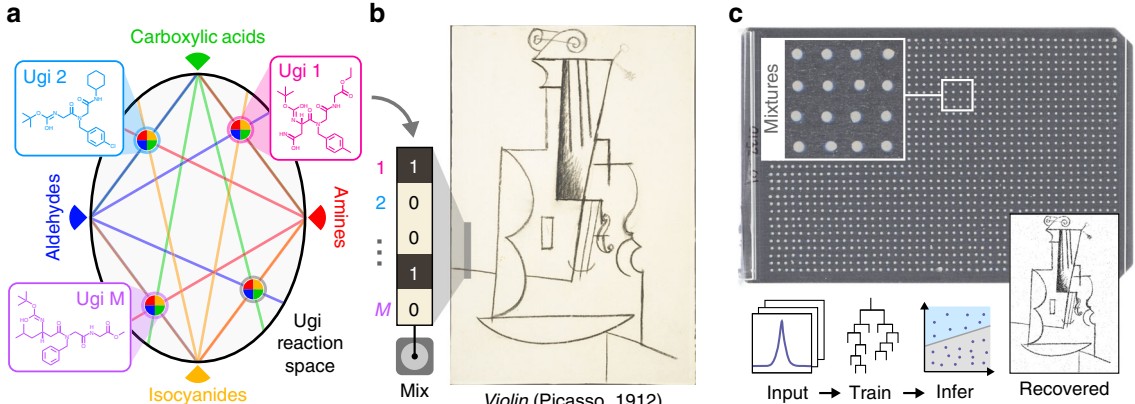

**Fig. 1 Process overview. a** Multicomponent reactions such as the four-component Ugi reaction can generate diverse libraries of small molecules. **b** A file can be stored by mapping its digital information onto a set of vectors which determine the presence ("1") or absence ("0") of multiple Ugi products at each location in an array. Here, we encoded a Cubist charcoal drawing of a violin by Pablo Picasso[19] (©Estate of Pablo Picasso/Artists Rights Society (ARS), New York). **c** The molecular data is stored on a thin metal plate in small crystalline spots. The original file can be recovered by interrogating each spot with mass spectrometry and interpreting the spectra with models trained to identify library elements. The spots on the data plate displayed here have an average diameter of 820 μm and a thickness of several micrometers.

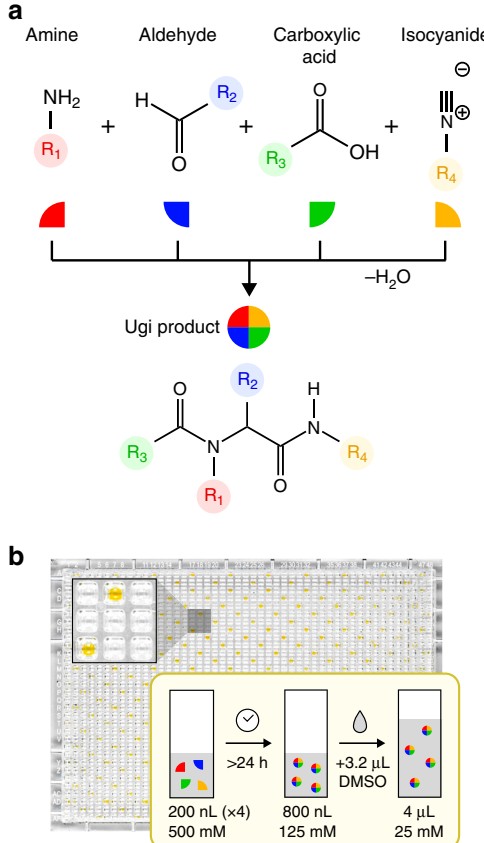

**Fig. 2 Library synthesis. a** The multicomponent Ugi reaction incorporates an amine, aldehyde, carboxylic acid, and isocyanide into a single peptide-like bis-amide. **b** We use automated acoustic liquid handling to synthesize combinatorial Ugi libraries of up to 1500 compounds at a time.

up to 1500 bits of information per mixture. However, using the full library in this way would strain our current experimental system, requiring large mixture volumes and low analyte concentrations. Thus when using this simple encoding scheme we often prefer to use a reduced subset of library components.

Figure 4 shows a 48,841-pixel binary image of the Egyptian god Anubis, which we encoded across 1527 independent mixtures using a 32-product library subset. In this case, the binary image data is first rearranged as a 1527 × 32 matrix, where each row of the matrix corresponds to one location on a data plate and each column corresponds to one library component. At each position, if a library element is meant to be included, we instruct our acoustic liquid handler to dispense a 2.5 nL droplet from its library well to the data plate. If it is meant to be excluded, no transfer is performed. The data is assembled on a standard MALDI target plate, forming a unique data mixture at each position. Finally, MALDI matrix is added to the mixtures, which are then dried, leaving behind crystalline spots, which can be stored and later read back using mass spectrometry.

We often create up to 1536 unique mixtures per data plate, and the storage capacity scales with the number of library compounds used. To encode the 0.88 megapixel image of a Picasso drawing shown in Fig. 1, we used 575 unique compounds. Additional storage experiments are summarized in Fig. 6 and Supplementary Fig. 7. These chemical datasets take several hours to write and read, are stable for at least 9 months, and can be read more than 100 times (Supplementary Figs. 8 and 12).

**Reading data from chemical mixtures**. To recover a chemical dataset, we analyze each mixture with mass spectrometry and train supervised learning algorithms to identify which library elements they contain. In the simplest version of this analysis, we can consider only the sodiated peaks of each Ugi product. If a peak's intensity exceeds a threshold, we record the compound as present ("1") and otherwise declare it as absent ("0"). For example, the spectrum shown in Fig. 4d contains the 1st, 2nd, 5th, 12th, 14th, and 17th compounds from a 32-compound sub-library, and thus this mixture encodes the following four bytes: 11001000 00010100 10000000 00000000.

Using only sodiated product peaks, the Anubis dataset was recovered with 97.9% accuracy. At least 30/32 compounds were correctly assigned in over 95% of mixtures and the residual errors displayed similar rates of false positives and false negatives (Fig. 4c). While the majority of compounds achieved <5% error, we observed an order of magnitude variation in compound performance (Fig. 4c). We can improve the readout accuracy by using multiple spectral features to determine the presence of each

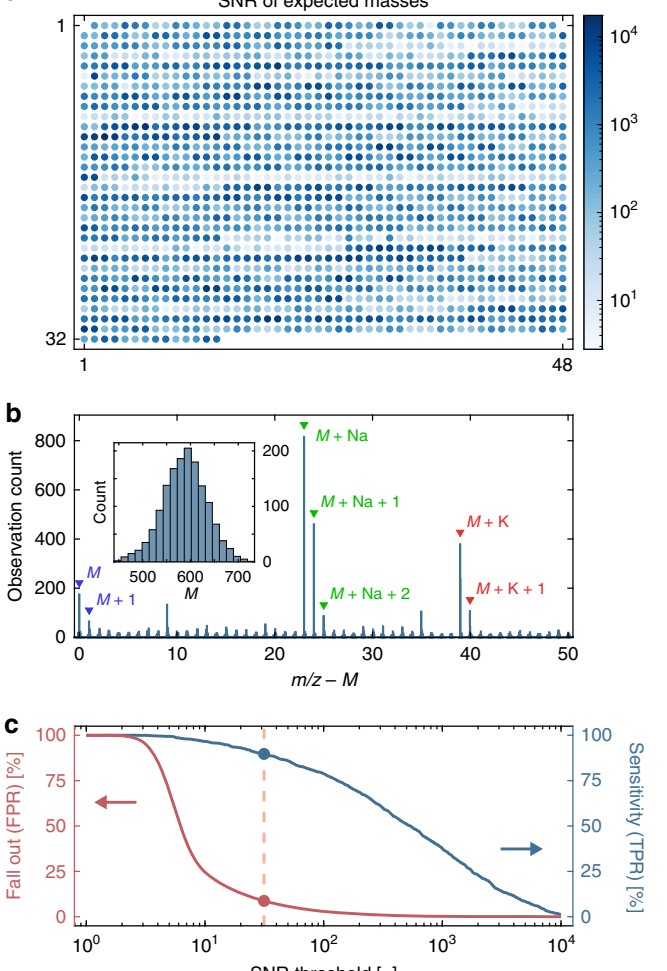

**Fig. 3 Library analysis. a** A color-coded map of the signal-to-noise ratio (SNR) of sodiated product peaks across a 1500-Ugi library. **b** A histogram of the 50 strongest peaks in each product spectra, offset by their expected Ugi product masses (M). Several common salt adducts and isotopes are labeled. Inset: A histogram of the monoisotoptic masses of the 1500-compound library. **c** The true positive rate (TPR) and false positive rate (FPR) as a function of the SNR threshold for the M+Na peaks. The dashed line (SNR ≈ 31) jointly optimizes the true and false positive rates ( TPR = 89.73%, FPR = 8.71%).

library element. Although it is difficult to assign the precise origin of every peak in a spectrum (e.g., isotopes, adducts), knowing their physical origin is not strictly required for data recovery. In fact, any feature which reliably correlates with the presence of a compound can be used to improve detection.

In order to exploit these correlated background peaks, we use supervised learning to train classifiers on library elements using logisitic or random forest regression. In both cases, we produce a set of regression models, one per library element, which can interpret the contents of a data mixture from its mass spectrum. By applying these machine learning approaches, we have achieved up to a five-fold reduction in error rates (Fig. 5).

In addition to improving accuracy, treating the readout as a learning problem has also revealed some interesting and non-intuitive chemical identifiers (Supplementary Fig. 13). For example, some classifiers used information about other library elements, perhaps as a result of competitive ionization. In other cases, compounds were found to form complexes with residual starting reagents, which could have arisen during ionization or during

synthesis. Moreover, by learning these difficult-to-anticipate interactions, multi-peak detection can allow us to identify multiple library elements with the same monoisotopic mass.

**Improving accuracy with sparse data-mixture maps**. For the Picasso drawing (Fig. 1) and Anubis image (Fig. 4), each bit of data was independently mapped onto the presence or absence of a single compound in a single mixture. While conceptually straightforward, this mapping limits the design parameters available to optimize experimental throughput and accuracy. For example, it is vulnerable to errors if single chemical components are improperly identified, and it implies that increasing storage capacity per spot requires both larger mixtures and larger libraries.

In Fig. 5, we explore an alternate encoding scheme, in which a 16-bit block of data is mapped to an entire mixture, instead of mapping each bit independently. This approach allows us to tune the complexity of the mixtures separately from the library size. Here we use a library subset of 512 compounds, but constrain exactly 32 compounds to be present in each mixture. In theory, there are $\binom{512}{32} \approx 2^{169}$ such combinations. However, only $2^{16}$ states are needed to represent all possible values of the 16-bit data. This sparse mapping implies that the vast majority of possible mixtures should never be observed. As such, when errors do occur, data can be rounded to the nearest valid mixture, providing some degree of fault tolerance. In this example, the minimum Hamming distance between any two valid mixtures is 36, meaning that perfect recovery of the encoded data can be guaranteed even when up to 17 of the 512 compounds (3.3%) are incorrectly classified. To test this, we performed a series of simulations where a 1600-bit data vector was encoded into 100 chemical mixtures. The virtual mixtures were symmetrically corrupted, at various error rates, and then decoded (Fig. 5b). Even with raw error rates several times larger than the guaranteed threshold, the vast majority of errors could still be corrected.

Using this sparse encoding, we wrote a 24,336-pixel digital image derived from a 16$^{th}$ century German illustration of angels seated at a table[40]. Looking only for the sodiated product peaks, we correctly classified 389 of the 512 library components, and after rounding to the nearest valid mixture, the original data was recovered with 96.67% accuracy (Fig. 5c). Training a logistic regression model to perform multi-peak detection resulted in a 5-fold reduction in raw compound errors (Fig. 5d) and a 30-fold reduction in decoded data errors, yielding a final accuracy of 99.89% (Fig. 5e).

Several mapping and detection schemes were tested in this work, and the results summarized in Fig. 6 highlight key trade-offs between the different approaches. The largest file was stored using direct mapping, since it provides a direct scaling of data capacity with library size. In contrast, the lowest error rates were achieved with sparse mapping and multi-peak detection.

## Discussion

In this study, we have introduced chemical information representations based on mixtures of multicomponent molecules. We can view this as an effort to store information in a superset of the molecular space available to biological systems, where synthetic chemistry is not held to the same environmental and energetic constraints as living cells. The demonstrations presented here are already six times larger than the information capacity of the smallest known genome[41], and although it is difficult to quantify exactly how much information is represented in living systems, it is interesting to think about how engineered chemical

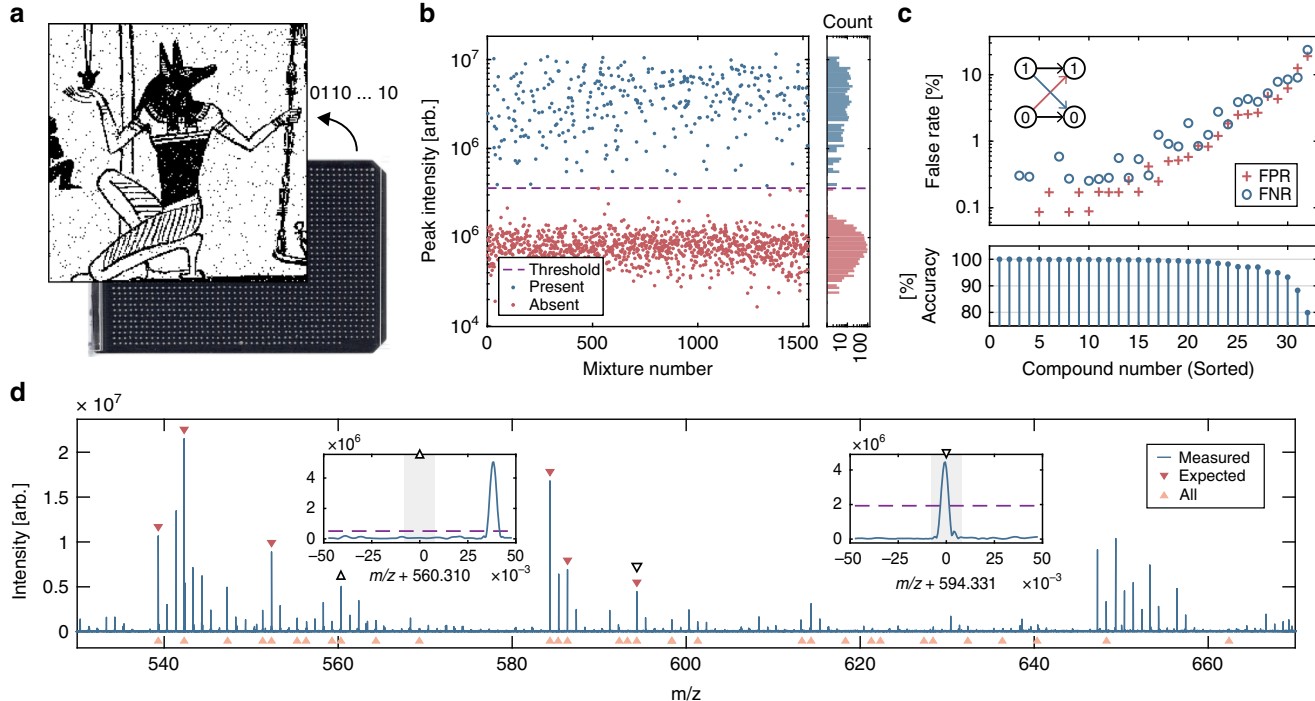

**Fig. 4 Encoding data into small-molecule mixtures. a** This molecular dataset is a 48,841 pixel (221 × 221) binary image of the Egyptian god Anubis[52]. **b** Sodiated peak intensities for one library element across the 1527 mixtures. **c** False negative and false positive rates for the 32 compounds, ordered by overall accuracy. **d** The mass spectrum of a data mixture containing six molecules from the 32-compound subset of the Ugi library. Left inset: the 8th molecule is absent ("0"). Right inset: the 17th molecule is present ("1").

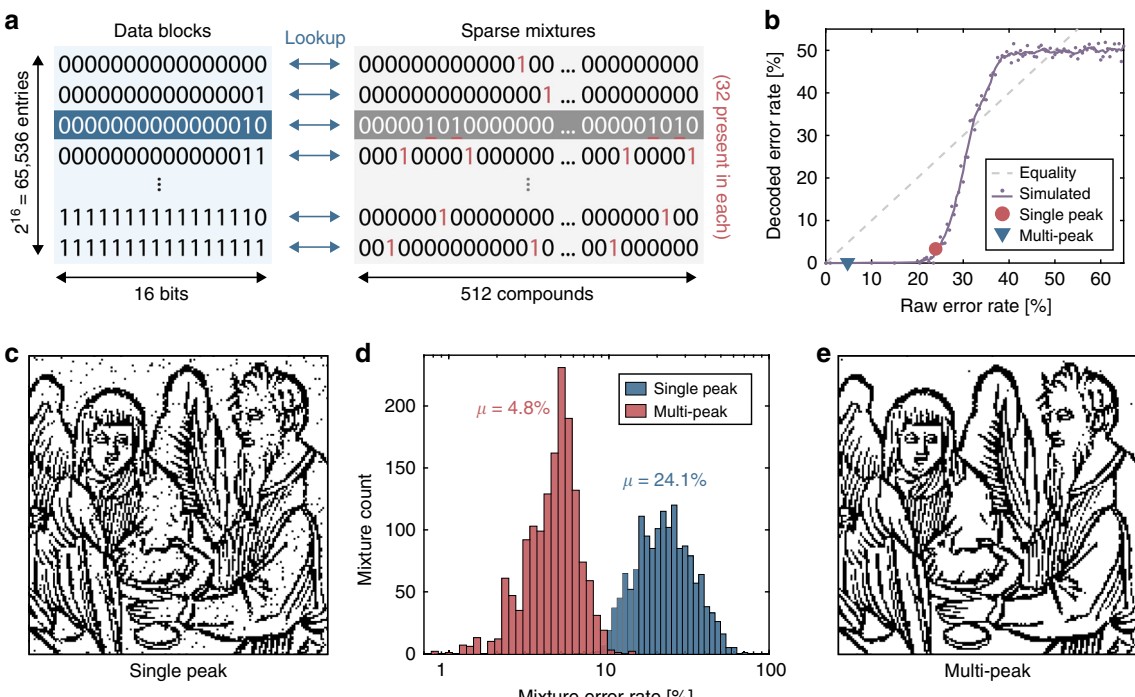

**Fig. 5 Using sparse mixture mapping and multi-peak readout to improve data recovery. a** Every 16 bits of data is mapped onto a sparse mixture, based on a 512-Ugi library subset, in which only 32 of the 512 of the compounds are present. There are ~$2^{169}$ such mixtures, but only $2^{16}$ are mapped to data. **b** Simulated read error rates, before and after decoding. Experimental results from single peak (**c**) and multi-peak (**e**) detection are also shown. **c** A 24,336 pixel binary image of angels at a dinner table from a 16th century print[40], recovered with single (sodiated) peak readout (96.67% accurate). **d** Histograms of the raw compound errors per mixture for each recovery scheme. **e** The digital image recovered with multi-peak detection (99.89% accurate).

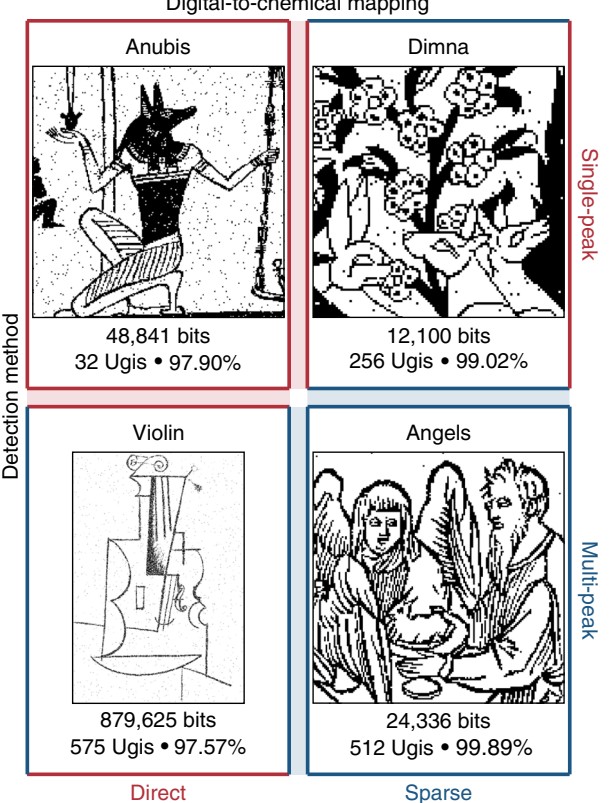

Digital-to-chemical mapping

**Fig. 6 A gallery of digital images written into mixtures of Ugi products.** The file sizes, number of compounds used, and readout accuracies are shown below each recovered image. Each dataset was represented using ~1500 mixtures. The Violin experiment was performed twice with similar results. The images were adapted from artwork from The Metropolitan Museum of Art (Anubis[52], Dimna[53], Angels[40]) and[19] the ©Estate of Pablo Picasso/Artists Rights Society (ARS), New York (Violin[19]).

information systems could similarly take advantage of the interplay between macromolecules and small molecules.

By introducing automated synthesis and analysis approaches for multicomponent Ugi products, we produced the largest small-molecule digital information representations described to date. We showed that by using sparse data-to-mixture mapping and applying supervised learning to mass spectrometry data, we can tolerate impurities, improve accuracy, and produce a workflow that readily generalizes to other classes of small molecules. We previously demonstrated data storage using a library of common metabolites[14], which could also benefit from the improved encoding and analysis strategies developed here. However, metabolite memory is more challenging to scale and perhaps more suitable for transient memory or chemical computation. Ugi products, on the other hand, offer combinatorial scaling, excellent stability, and comparatively uniform chemical properties.

In total, we stored 1.8 million bits of data in Ugi molecules, including more than 0.8 million bits on a single plate (Fig. 6). This is still very far from theoretical capacity limits, and there are many parameters that can be further optimized, including library size, plate spot density, and mixture complexity. With incremental improvements in these areas, we could reach several megabytes per plate or more, using largely the same experimental setup. Introducing photolithography or advanced printing could further improve the spatial density and data capacity of a plate by several orders of magnitude. We selected steel plates for their

compatibility with available instruments, but future implementations could utilize flexible substrates or reels, or embed chemical information onto the surfaces of three-dimensional objects. MALDI imaging can achieve resolutions finer than 10 microns[42], and although reading larger chemical datasets may require different coding strategies, we have not yet approached the information capacity of our readout.

The sensitivity of MALDI mass spectrometry is limited by the presence of background ions, such as matrix adducts and impurities[43]. With adjustments to ionization, trapping, and excitation, this chemical noise can be mitigated, enabling attomole limits of detection[44,45]. In our current demonstrations, we estimate the amount of Ugi product ionized per read to be on the order of 10 femtomoles (Supplementary Fig. 12), offering room for future improvement.

There are also other interesting avenues to explore beyond simple capacity improvements. Our sparse mixture mapping (Fig. 5) can be considered a coarse version of block coding, and there would be benefits to exploring more efficient coding schemes for digital error correction. Alternatively, one could leverage sparsity for enhanced information density, and represent many bits per small molecule present. The experimental workflow used here has similarities with early-stage pharmaceutical pipelines[46], and it would be exciting to consider how error correction and the correlated statistics of encoded mixtures could be applied to drug discovery and medical applications.

By automating the Ugi reaction, we found that we could synthesize thousands of multicomponent compounds per day using low cost reagents and with minimal manual sample preparation. The yield and quality of the Ugi reactions supported decoding hundreds of bits of data per mixture without any purification, which is partly a result of the fact that here we are interested in the information carried by the unique fingerprint of each library element rather than individual chemical structures. One minor reaction adjustment that we found helpful was to limit the amount of isocyanide to 80% of the other reagents, which seemed to reduce side product formation.

Our ability to apply supervised learning to chemical information recovery stems from the availability of labeled training data, and a willingness to tackle complex mixtures. To utilize the potential of even larger molecular libraries, other approaches may be required. Recent studies have explored the use of autonomous systems for the exploration of chemical spaces[47], which pairs well with the idea of mapping chemical mixtures to abstract information. Screening combinatorial libraries in bulk rather than one at a time is already established in some areas of molecular biology, such as aptamer design[48]. Extending these information-centric philosophies to more subtle molecular properties and emergent chemical reaction networks may prove particularly fruitful.

## Methods

**Materials and reagents.** The solvent dimethyl sulfoxide (DMSO, anhydrous, ≥99.9%, MilliporeSigma) was used to prepare all solutions in the library and data plates. Analytical grade α-cyano-4-hydroxycinnamic acid (HCCA, ≥99.0%, MilliporeSigma) was used as the matrix material for all MALDI samples. The library of 1500 Ugi products was constructed with the following five amines: benzylamine, 4-methylbenzylamine, p-methoxybenzylamine, 4-chlorobenzylamine, 4-tertbutylbenzylamine; five aldehydes: cyclohexanecarboxaldehyde, 3-cyclohexylpropanal, valeraldehyde, isovaleraldehyde, cyclopentanecarboxaldehyde; 12 carboxylic acids: Boc-glycine, Boc-proline, Boc-N-methyl-L-valine, Boc-L-asparagine, Boc-L-beta-homoleucine, Boc-L-methionine, Boc-L-beta-homoglutamine, Boc-L-beta-homo-methionine, Boc-L-phenylalanine, Boc-N-alpha-N-epsilon-formyl-L-lysine, Boc-N-methyl-L-phenylalanine, Boc-O-methyl-L-tyrosine; and five isocyanides: cyclohexyl isocyanide, ethyl isocyanoacetate, benzyl isocyanide, 2-naphthyl isocyanide, methyl isocyanoacetate. These compounds were obtained at synthesis grade or higher and used as received from their vendors (Chem-Impex for the carboxylic acids and MilliporeSigma for the others). Further details about the reagents can be found in Supplementary Fig. 1.

**Library preparation**. Each reagent was dissolved in DMSO to a concentration of 500 mM and placed into a 384-well plate. Using an acoustic fluid handler, we dispensed the reagents, 200 nL per inclusion, into a 1536-well plate to enumerate all possible four-component Ugi reactions. The array of reagent mixtures was left to react at room temperature for 1–2 days. After reacting, DMSO was added to each library well to reach a final volume of 4 μL.

**Mass spectrometry**. Mass spectra were acquired with a Fourier transform ion cyclotron resonance (FT-ICR) mass spectrometer in positive ion mode. Samples were crystallized in matrix (Supplementary Fig. 17), using an ~100:1 ratio of matrix to Ugi product. Samples were ionized using matrix-assisted laser desorption ionization (MALDI). Spectra produced by FT-ICR are particularly high resolution, often reaching peak widths of 0.001 Da or smaller. To ensure the accuracy of peak assignment, a mass calibration is performed before each run using sodium tri-fluoroacetate as a reference[49] (Supplementary Fig. 10). We typically acquire spectra for 1.5 s, which results in a resolving power of $1.3 \times 10^5$ at 600 Da (Supplementary Fig. 15). The instrument serially addresses each crystallized spot (Supplementary Fig. 11), and takes about 4 h to record all 1,536 spots on a plate. Each measurement is made by ionizing a portion of a sample with a laser configured to take 500 shots at 1000 Hz, over a scan area of 500–900 μm, with medium focus, and ×4 averaging. We convert the raw data files from the instrument into a custom HDF5 file, for more efficient querying and ease of access. To normalize signals across measurements, we often convert the raw intensity values of a spectrum to signal-to-noise ratios (SNR) according to the following shift-and-scale relation:
$\mathrm{SNR} = (I - \mu)/\sigma$, where $I$ is an intensity and $\mu$ and $\sigma$ are the mean and standard deviation of the spectrum's background (see Supplement).

**Library validation**. To identify successful reactions, a small volume (20 nL) from each library well was spotted to a unique location on a stainless steel plate (78 mm × 120 mm) along with matrix (20 nL of 176.2 mM HCCA in DMSO). The plated samples were allowed to dry overnight (~10 h) into round crystals (~800 μm in diameter), before analysis via mass spectrometry. In the resulting mass spectra, we looked for peaks corresponding to expected Ugi product masses, and used peak height as a coarse measure of reaction yield. Since the Ugi products have similar ionization profiles, we performed a global statistical analysis of the library spectra, using the SNR of their sodiated peaks. A common threshold ($\tau$) was found using receiver operator characteristic (ROC) curve analysis[50]. To construct the ROC curve, we look for the sodiated product peaks across all reaction wells, apply a given SNR threshold to assess the presence or absense of these peaks, tally detected library peaks to estimate the true positive (TPR) and false positive (FPR) rates, and repeat this process for all candidate thresholds. Since there should be exactly one product per well, if the expected product is detected, it is counted as a true positive (TP), and if not, then it is marked as a false negative (FN). Similarly, if other products are detected in the well, they are counted as false positives (FP) and otherwise as true negatives (TN). The products with masses that overlap with that of the expected product are counted as TPs or FNs. Error rates can be calculated as $\mathrm{TPR} = \mathrm{TP}/(\mathrm{TP} + \mathrm{FN})$ and $\mathrm{FPR} = \mathrm{FP}/(\mathrm{FP} + \mathrm{TN})$, and used to find an optimal SNR threshold, by minimizing the distance to the (0,1)-corner:
$\left[ (0 - \mathrm{FPR}(\mathrm{SNR}))^2 + (1 - \mathrm{TPR}(\mathrm{SNR}))^2 \right]^{1/2}$. The Ugi products whose SNR exceeds this threshold ($\mathrm{SNR} \geq \tau$) are declared present.

**Data plate preparation**. First, a digital file is converted into a one-dimensional binary vector. This vector is then encoded, either with a direct or sparse mapping, into an $M \times N$ compound-presence matrix, where $M$ is the number of compounds to be used, and $N$ is the number of independent mixtures to be made. The value of element $p_{mn}$ in this matrix indicates the presence ("1") or absence ("0") of the $m^{th}$ compound in the $n^{th}$ mixture. To physically generate the mixtures, 2.5 nL droplets are transferred from the 1536-well library plate to their appropriate locations on a MALDI plate. Finally, 30 nL of matrix solution (176.2 mM HCCA in DMSO) is added to each data mixture spot. The overall time to write a data plate ranged from 0.3 to 7.9 h, varying with the encoding scheme and file size (Supplementary Figs. 6 and 7). Once all transfers are complete, the data plate is left to dry in a fume hood overnight or a vacuum chamber for about 2 h. The resulting dried mixture spots are typically 1 mm in diameter. Currently, the number of compounds that can be included in each mixture is limited by the layout of samples on a MALDI plate. For a 1536-well grid, spots can contain up to 200 nL of solution before they begin to merge with adjacent samples (Supplementary Fig. 14). For more complex samples, mixing would have to be done in an intermediate well plate.

**Data plate analysis**. During plate preparation, the matrix solution is spiked with a reference Ugi molecule (Supplementary Fig. 5) which is used to calibrate for small offsets in the recorded masses. After offset calibration, raw mass spectra are resampled to a common m/z grid in order to construct a single analysis-ready matrix containing the mass spectra of all spots on a plate.

For single peak detection, the sodiated adduct intensities for a product are simply one row in the spectral matrix, and this vector can be thresholded to determine, which mixtures contain the compound. The detection threshold for each compound was found using ROC analysis of labeled training data, as

previously described for library validation. Recovering the data file from the presence matrix depends on the encoding method. For direct mapping, the matrix is simply reshaped to obtain the stored data. For sparse mappings, each matrix row was matched to the nearest valid key and converted to the corresponding binary data value.

For multi-peak detection, a similar procedure was followed, except that the presence matrix was found by applying a regression model trained to identify each compound based on multiple spectral features. To reduce computational overhead, instead of building the models on the entire mass spectra matrix, masses whose average intensities were close to the noise floor were discarded, reducing the feature space to <1% of its original size, from four million initial points to at most 20,000 candidate masses. For logistic regression, these features were further refined based on AUROC scores. This additional filtering was not needed for random forest regression since it automatically performs feature selection. The Python library Scikit-learn[51] was used to construct a regression model for each compound. Logistic regressions were configured to use 64 spectral peaks, while random forest regressions were configured to use 300 trees of unlimited depth and at most 20,000 spectral features. The regression models used a 30/70 train/test split.

## Data availability
The datasets from this study are available from the authors on reasonable request.

## Code availability
The software used in this study is based on code available from the Metabolomics Workbench data repository (study ST001173), and is available from the authors on reasonable request.

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

## Acknowledgements

This research was supported by funding from the Defense Advanced Research Projects Agency (DARPA W911NF-18-2-0031). The views, opinions, and/or findings expressed are those of the authors and should not be interpreted as representing the official views or policies of the Department of Defense or the U.S. Government. This work was also made possible by support from the Office of the Vice President for Research at Brown University, and by the National Science Foundation under Grant No. 1941344.

## Author contributions

C.E.A., E.K. and J.G. performed experiments. C.E.A., E.K., J.G. and J.K.R. analyzed data. C.E.A., A.D., K.O., S.-L.C. and L.S. synthesized the library. J.S., P.M.W., S.R., C.R., M.O., E.K., B.M.R. and J.K.R. provided direction and oversight. C.E.A., E.K. and J.K.R. drafted the paper. All authors provided notes and edits to the paper.

## Competing interests

A pending patent application (PCT/US2019/038301) has been filed by Brown University with the following authors included on it as inventors [C.E.A., S.L.C., A.D., J.G., E.K., E.K., K.O., S.R., C.R., J.S., P.M.W., B.M.R., and J.K.R.] concerning data storage and computation using small molecules including Ugi reaction products.
