## [Peer Review File · Nature Communications]

Reviewers' Comments:

Reviewer #1:

Remarks to the Author:

This article by Rosenstein and coworkers describes the preparation and utilization of a memory device involving mixtures of low molecular weight components. In particular, the authors took advantage of the well-known multicomponent Ugi reaction to create combinatorial libraries of small molecules with unique molecular signatures. Afterwards, specific combinations of molecules were spotted on a data plate. Each spot on the data plate is then decoded by MALDI FT-ICR mass spectrometry. The presence or absence of a given molecule in a particular spot is correlated to a binary sequence, thus allowing encryption of pixelized images. Altogether, the scientific quality of this manuscript is excellent. The reported concept is clever and could certainly be of interest for the broad readership of Nature Communications. However, prior to publication, the authors shall consider the following remarks:

1- Introduction: in their first sentence, the authors have only mentioned DNA as a sequence-defined macromolecule enabling data storage. They omitted to mention that several abiotic information-containing polymers have been reported in recent years. This aspect of the field shall be included. For instance, overviews on the topic (Lutz/Colquhoun, Nature Chemistry 2014; Nolte, Nature Reviews Chemistry 2018) as well as specific examples published in Nature Communications (Lutz, 2015, 2017, 2019; Du Prez, 2018 (only listed in SI); Zhang, 2019) shall be cited.

2- Figure 2: only shows a general view of the formed Ugi products. There is unfortunately no information in the main text about the chemical nature of the substituents R1-R4 (displayed only in Figure S1). I suggest to transfer this important information (or at least some examples of it) in Figure 2.

3- By displaying all the formed molecules very small, Figure S3 gives a good impression of the broadness of the investigated molecular space. However, what would be also very important for the readers is to have a table listing and comparing the molar mass of all formed Ugi products. From the reactants shown in Figure S1, it appears that the molar mass of some products formed might only differ from a Da or less than a Da. Although the reading error rates were obviously minimized in this work, the authors shall discuss the MS identification of structurally-close products. This could be done by displaying in the SI MS spectra of model mixtures of closely-related products.

4- In the first two sections of the main text about writing and reading, it would be important to include clear information about the overall reading and writing times. In other words, how long does it take (in seconds, minutes or hours) to prepare a coded data plate using a liquid handler. Same question for the MS analysis. What is the duration of a comprehensive data plate analysis?

5- The diameter of a single spot shall be mentioned in the main text and in the methods section.

6- Page 4: costs in dollars are unnecessary for a scientific publication. The authors may only mention that the system is potentially cheap.

Reviewer #2:

Remarks to the Author:

*** Summary:

This paper discusses the use of multicomponent molecules, embodied as molecules resulting from four-component Ugi reactions, for digital data storage. Digital data is represented by mixes of such molecules, which are synthesized and mixed robotically with an acoustic handler. The molecular

mixes are read via mass spectrometry and machine learning is used to map resulting signals back to the bits that originated them. Machine learning obviates the need to understand side chemical reactions that may happen after the mixtures are created, as it provides a direct mapping from signals to bits. The paper also investigates the benefits of using a sparse mix (i.e., using a subset of all available four-component molecules and mapping mixes to a set of bits) with machine-learned multi-peak detection versus using a dense mix (i.e., using all available four-component molecules and mapping each to a single bit) or even a sparse mix with machine-learned sodiated-product single-peak detection, and concludes that the former provides added signal detection and error correction capability.

*** Contributions:

The novel portions of this work compared to prior work are:

1) Use of four-component Ugi molecules for digital data storage: Molecular mixes created robotically had been proposed before by the authors, as appropriately cited and included among the references. That work used molecules picked from the metabolome, with a one-to-one mapping to bits within a mix. This meant that growing the number of bits within a mix relied on finding a proportional number of new molecules compatible with the system. In this work, the combinatorial nature of four-component Ugi molecules amplifies the bit representation power of a new subcomponent molecule because it can be associated with all combinations of the other subcomponents.

2) Use of a sparse mix of molecules associated with machine learned multi-peak detection for digital data storage: Sparse representations are common in coding theory and are used when the storage/communication channel is noisy. In specific, this paper applies linear block coding concepts to molecular mixes, and shows that is it more effective at correcting errors when using machine learning models trained for multi-peak detection. These models work better because they are tolerant to interactions between different molecules in a mix.

Both these observations are novel and interesting. Additionally, the ability to automate multi-component Ugi reactions and create molecular mixes with them may also be interesting to scientists working on high-throughput screening of small molecules. I wish the authors would elaborate on this, as it would make the paper interesting to a broader set of readers, and could spark collaborations and new areas of work.

*** Methodology/evidence/argument soundness:

This paper does provide sufficient supporting evidence for the two novel portions described above.

My main issue with this paper is that much space is dedicated to arguing for molecular mixes as a valid approach for digital data storage, and not much emphasis is placed on clearly delineating and demonstrating the value of what is truly novel in this paper. To be clear, this is more of a presentation problem rather than a technical issue.

I wish the text more clearly emphasized what is new and more thoroughly studied the effect of the multiple techniques being explored. Figure 6 is an attempt at showing the tradeoffs between the direct/sparse encodings and the single-peak/multiple-peak detection approaches, but it could be a lot more detailed. The text also covers the benefits of Ugi molecules versus metabolomes, but since this is one of the new topics covered in this paper, I would have expected more emphasis on it as well.

A few comments about the contents of the paper that could be improved:

- The paper claims the absence of amplification of molecules as a positive feature. This statement

is misleading. The ability of amplification of molecules is actually desirable in data storage because it may improve the storage density (store only a few molecules of each type and amplify before reading to increase the signal) and also facilitate replication and distribution of data.

- Some of the text seems to make claims about the density of a multicomponent molecular memory. However, it does not provide quantification of practical densities. For example, one paragraph in the introduction claims "encoding between 16 and 575 bits of data at a single memory location compares favorably to electronic systems, where multi-level flash memory cells are typically limited to four bits or less". Even though this statement is not false, it says nothing about the volume occupied by these "locations". A flash memory cell is much smaller than one of the multicomponent molecular memory locations presented in this paper. One might try to claim that once this technology scales to smaller locations it would be denser than flash memory, but this would require a discussion of the expected limits of scaling a multicomponent molecular memory, which is not satisfactorily provided in this paper.
- A general comment about the stability of the information in the mixes: it is commendable that the authors have attempted the experiments multiple times over the period of 9 months. It would be interesting to see results for accelerated aging experiments to assess how the technology would do longer term. I do realize that this is beyond the scope of this paper, but it would be helpful to see it suggested in the text, along with expected results, and more comment on the effects of humidity, exposure to light, and different temperatures on the contents stored.
- I couldn't find an explanation for how the simulated curve in Figure 5b is generated. Either add one if not yet covered anywhere, or provide a pointer to where it is described.
- It seems laborious to repeat experiments, but it also seems like a single experiment per configuration in Figure 6 may not capture all relevant effects (although one of the configurations has 2 experiments). At the very least, the authors should provide an argument for why 1-2 experiments per configuration is a reasonable thing to do.
- It is not clear how far the multi-peak machine learning approach is expected to go. As the number of components in the sparse representation grows, how is the machine learning detection expected to scale?
- The parametric sweeps in the Supplement are interesting and their implications should be included in the discussion.
- The methods section could be a lot more detailed, and it should be self-contained. For example, it is missing how many Ugi components of each type are used (5x5x12x5), and how many reactions are sufficient for all possible combinations (1,500). The library validation description, along with how thresholds are set, is quite confusing. There is also no mention of whether this applies for the single-peak approach, multi-peak approach, or both. Finally, a lot more detail on how the machine learning portion of the work is trained and used for inference, in both single- and multi-peak approaches is needed for reproducibility.

*** Minor comments:

- Legend of Figure 1: provide size of small dried spots (including height).
- In page 8, it would be helpful to see examples of "difficult to anticipate relationships", mainly for non-chemists reading the paper. It seems appropriate for the Supplement.
- In supplementary information:
 - Mention in Figure S2f caption that the labels in the x axis are PubChem IDs.
 - I am not sure how useful Figure S3 really is at these sizes.
 - Figure S5c: it would be interesting to know why the left side of the curve is different from the rest.

- Figure S7: It would be helpful to label the different points in the plot and table with their reference/citation numbers.
- Supplementary section 3.4 would benefit from further discussion about how far technology can scale to smaller spots/less material. Or perhaps a more comprehensive discussion could follow Supplementary section 4.

Dear Reviewers,

Thank you for your supportive and constructive comments. Please find our responses below. Enclosed are revised versions of the manuscript and supplement with changes **highlighted in red**.

Sincerely,
Chris Arcadia and Jacob Rosenstein

Reviewer 1

This article by Rosenstein and coworkers describes the preparation and utilization of a memory device involving mixtures of low molecular weight components. In particular, the authors took advantage of the well-known multicomponent Ugi reaction to create combinatorial libraries of small molecules with unique molecular signatures. Afterwards, specific combinations of molecules were spotted on a data plate. Each spot on the data plate is then decoded by MALDI FT-ICR mass spectrometry. The presence or absence of a given molecule in a particular spot is correlated to a binary sequence, thus allowing encryption of pixelized images. Altogether, the scientific quality of this manuscript is excellent. The reported concept is clever and could certainly be of interest for the broad readership of Nature Communications.

However, prior to publication, the authors shall consider the following remarks:

1- Introduction: in their first sentence, the authors have only mentioned DNA as a sequence-defined macromolecule enabling data storage. They omitted to mention that several abiotic information-containing polymers have been reported in recent years. This aspect of the field shall be included. For instance, overviews on the topic (Lutz/Colquhoun, Nature Chemistry 2014; Nolte, Nature Reviews Chemistry 2018) as well as specific examples published in Nature Communications (Lutz, 2015, 2017, 2019; Du Prez, 2018 (only listed in SI); Zhang, 2019) shall be cited.

We thank the reviewer for their encouraging comments and helpful recommendations. We have added citations to Nolte's and Lutz's reviews and included several of the examples of non-biological polymer memory to the introductory paragraph.

2- Figure 2: only shows a general view of the formed Ugi products. There is unfortunately no information in the main text about the chemical nature of the substituents R1-R4 (displayed only in Figure S1). I suggest to transfer this important information (or at least some examples of it) in Figure 2.

This is certainly true, thank you for catching this. Figure 2(a) is meant to show the general form of the reagents and products, but we have added the full names of each reagent to the Methods, under a new “Materials and Reagents” section. Supplemental Figure S1 also lists all of these components.

3- By displaying all the formed molecules very small, Figure S3 gives a good impression of the broadness of the investigated molecular space. However, what would be also very important for the readers is to have a table listing and comparing the molar mass of all formed Ugi products. From the reactants shown in Figure S1, it appears that the molar mass of some products formed might only differ from a Da or less than a Da. Although the reading error rates were obviously minimized in this work, the authors shall discuss the MS identification of structurally-close products. This could be done by displaying in the SI MS spectra of model mixtures of closely-related products.

We appreciate this comment, but we feel showing the masses of all 1,500 compounds would be cumbersome to read. However, a histogram of the masses is shown in the inset of Figure 3.b, and a table of all of the starting reagents is provided in Fig S1.

It is true that some of the library products have overlapping monoisotopic masses. Our multi-peak detection can still distinguish among such products (through auxiliary peaks such as those from leftover reagents or contaminants), although since our write speed is not yet fast enough to utilize all 1500 library products in a single dataset, we have not needed to use this. We have added a new “Library Refinement” section to the supplement, explaining how one step in the library subset selection process is to select for well separated monoisotopic masses.

4- In the first two sections of the main text about writing and reading, it would be important to include clear information about the overall reading and writing times. In other words, how long does it takes (in seconds, minutes or hours) to prepare a coded data plate using a liquid handler. Same question for the MS analysis. What is the duration of a comprehensive data plate analysis?

Write time depends on the data file size and encoding, but our experiments ranged from 20 minutes to 8 hours. Read time depends on the number of data mixtures and the MS acquisition time (which affects m/z domain resolution). These parameters were more or less constant across our experiments, and the average read was about 3.2 hours. We have added a new supplementary section “Data Written”, which includes a table containing relevant details (including write and read time) for each data plate. We have also included the write and read

times of the Picasso dataset to the main text under the “Writing Data as Chemical Mixtures” section.

5- The diameter of a single spot shall be mentioned in the main text and in the methods section.

The spot diameter, along with the dimensions of the square inset of the plate in Figure 1.c, have been added to the Figure 1 caption. We have also added the average library spot diameter under Methods “Library Validation”. Similarly, the average Anubis data plate spot diameter was added under Methods “Data Plate Preparation”.

6- Page 4: costs in dollars are unnecessary for a scientific publication. The authors may only mention that the system is potentially cheap.

The estimated library cost has been removed and we have re-phrased this paragraph.

Reviewer 2

*** Summary:

This paper discusses the use of multicomponent molecules, embodied as molecules resulting from four-component Ugi reactions, for digital data storage. Digital data is represented by mixes of such molecules, which are synthesized and mixed robotically with an acoustic handler. The molecular mixes are read via mass spectrometry and machine learning is used to map resulting signals back to the bits that originated them. Machine learning obviates the need to understand side chemical reactions that may happen after the mixtures are created, as it provides a direct mapping from signals to bits. The paper also investigates the benefits of using a sparse mix (i.e., using a subset of all available four-component molecules and mapping mixes to a set of bits) with machine-learned multi-peak detection versus using a dense mix (i.e., using all available four-component molecules and mapping each to a single bit) or even a sparse mix with machine-learned sodiated-product single-peak detection, and concludes that the former provides added signal detection and error correction capability.

*** Contributions:

The novel portions of this work compared to prior work are:

1) Use of four-component Ugi molecules for digital data storage: Molecular mixes created robotically had been proposed before by the authors, as appropriately cited and included among the references. That work used molecules picked from the metabolome, with a one-to-one

mapping to bits within a mix. This meant that growing the number of bits within a mix relied on finding a proportional number of new molecules compatible with the system. In this work, the combinatorial nature of four-component Ugi molecules amplifies the bit representation power of a new subcomponent molecule because it can be associated with all combinations of the other subcomponents.

2) Use of a sparse mix of molecules associated with machine learned multi-peak detection for digital data storage: Sparse representations are common in coding theory and are used when the storage/communication channel is noisy. In specific, this paper applies linear block coding concepts to molecular mixes, and shows that is it more effective at correcting errors when using machine learning models trained for multi-peak detection. These models work better because they are tolerant to interactions between different molecules in a mix.

Both these observations are novel and interesting. Additionally, the ability to automate multi-component Ugi reactions and create molecular mixes with them may also be interesting to scientists working on high-throughput screening of small molecules. I wish the authors would elaborate on this, as it would make the paper interesting to a broader set of readers, and could spark collaborations and new areas of work.

We appreciate the reviewer's detailed comments. We have attempted to address the relevance to high-throughput screening with several added and expanded sections in both the main text and supplement.

*** Methodology/evidence/argument soundness:

This paper does provide sufficient supporting evidence for the two novel portions described above.

My main issue with this paper is that much space is dedicated to arguing for molecular mixes as a valid approach for digital data storage, and not much emphasis is placed on clearly delineating and demonstrating the value of what is truly novel in this paper. To be clear, this is more of a presentation problem rather than a technical issue.

I wish the text more clearly emphasized what is new and more thoroughly studied the effect of the multiple techniques being explored. Figure 6 is an attempt at showing the tradeoffs between the direct/sparse encodings and the single-peak/multiple-peak detection approaches, but it could be a lot more detailed. The text also covers the benefits of Ugi molecules versus metabolomes, but since this is one of the new topics covered in this paper, I would have expected more emphasis on it as well.

Thank you for these suggestions. We have added a more detailed comparison table of the data storage experiments to the Supplement, as well as a new paragraph describing the tradeoffs in

the main text (prior to the Discussion). We have also included a paragraph in the discussion comparing metabolite and multicomponent information systems.

A few comments about the contents of the paper that could be improved:

- The paper claims the absence of amplification of molecules as a positive feature. This statement is misleading. The ability of amplification of molecules is actually desirable in data storage because it may improve the storage density (store only a few molecules of each type and amplify before reading to increase the signal) and also facilitate replication and distribution of data.

This is a very fair comment. We have updated this sentence of the manuscript.

- Some of the text seems to make claims about the density of a multicomponent molecular memory. However, it does not provide quantification of practical densities. For example, one paragraph in the introduction claims "encoding between 16 and 575 bits of data at a single memory location compares favorably to electronic systems, where multi-level flash memory cells are typically limited to four bits or less". Even though this statement is not false, it says nothing about the volume occupied by these "locations". A flash memory cell is much smaller than one of the multicomponent molecular memory locations presented in this paper. One might try to claim that once this technology scales to smaller locations it would be denser than flash memory, but this would require a discussion of the expected limits of scaling a multicomponent molecular memory, which is not satisfactorily provided in this paper.

We have updated this paragraph in the introduction. Highlighting the number of bits at a single memory location is meant to emphasize how molecular memory utilizes new information dimensions to realize new achievements; storing >100 bits in a single flash memory cell would of course be impossible.

Regarding the potential for improved information density, of course we do feel that this is an important feature. As the reviewer correctly points out, our experiments are currently at millimeter scales. We do not have the instrumentation available to do a complete study of the true experimental scaling limits. Some of the newly added Supplementary discussion about repeated reads indirectly addresses part of this question; however this is all contingent on the performance of our specific MALDI mass spectrometer.

- A general comment about the stability of the information in the mixes: it is commendable that the authors have attempted the experiments multiple times over the period of 9 months. It would be interesting to see results for accelerated aging experiments to assess how the technology would do longer term. I do realize that this is beyond the scope of this paper, but it would be helpful to see it suggested in the text, along with expected results, and more comment on the effects of humidity, exposure to light, and different temperatures on the contents stored.

Thank you. Accelerated aging tests are on our to-do list, but we have not performed these. We have expanded the “Long Term Storage” section of the supplement and added some mention of parameters that could affect the chemical data stability.

- I couldn't find an explanation for how the simulated curve in Figure 5b is generated. Either add one if not yet covered anywhere, or provide a pointer to where it is described.

This is a good suggestion. We have added this explanation to the “Improving Accuracy with Sparse Data-Mixture Maps” section. For the simulated curve, a random 1,600-bit binary vector was written, *in silico*, to a mock data plate, with the same mapping that was used for the Angels dataset. The resulting 100 mixtures were corrupted at various bit error rates and then decoded to determine the accuracy of recovered data.

- It seems laborious to repeat experiments, but it also seems like a single experiment per configuration in Figure 6 may not capture all relevant effects (although one of the configurations has 2 experiments). At the very least, the authors should provide an argument for why 1-2 experiments per configuration is a reasonable thing to do.

We have added a comment about repeated experiments prior to the “Discussion” section. The two mappings (direct vs sparse) and two detection schemes (one vs many peak) yield quite different results, and we feel single experiments are enough to highlight some important differences in their fidelity and capacity. These differences are largely explained through the math of their encoding. Between our earlier work with metabolites, the work described in this manuscript, and other ongoing extensions with comparable datasets, we have performed quite a few experiments. Of course there is a bit of variability, but the encoding scheme has a large effect on the fidelity and capacity.

- It is not clear how far the multi-peak machine learning approach is expected to go. As the number of components in the sparse representation grows, how is the machine learning detection expected to scale?

We agree, this is a very interesting question. We are not yet near the limit. However, as the size of the library grows, the amount of required training data similarly increases. We can use sparsity to increase the storage capacity, but the supervised learning still requires that we have enough observations of each compound. The finite number of samples and the finite fluid handling speeds make it challenging for us to explore these limits experimentally.

- The parametric sweeps in the Supplement are interesting and their implications should be included in the discussion.

Thank you. We agree and have included the key-takeaways from these sweeps in the discussion and methods section.

- The methods section could be a lot more detailed, and it should be self-contained. For example, it is missing how many Ugi components of each type are used (5x5x12x5), and how many reactions are sufficient for all possible combinations (1,500). The library validation description, along with how thresholds are set, is quite confusing. There is also no mention of whether this applies for the single-peak approach, multi-peak approach, or both. Finally, a lot more detail on how the machine learning portion of the work is trained and used for inference, in both single- and multi-peak approaches is needed for reproducibility.

We appreciate this suggestion, and we have expanded the Methods. We have added a "Materials and Reagents" section where we list the number and names of the Ugi reaction reagents used to make a library. We have rewritten much of the "Library Validation" section, and clarified the thresholds in the "Data Plate Analysis" section. We have also expanded the description of the multi-peak approach including the machine learning details.

*** Minor comments:

- Legend of Figure 1: provide size of small dried spots (including height).

The diameter of the small dried spots has now been added to the legend. Dried spot height has been coarsely measured, using in-plane microscope imaging, to be about 10 micrometers (cited for a spot X24Y02, on the plate shown in Figure 1). However, we note that only a small fraction of the material in a spot is removed by the MALDI laser, which is the reason that the data can be read multiple times.

- In page 8, it would be helpful to see examples of "difficult to anticipate relationships", mainly for non-chemists reading the paper. It seems appropriate for the Supplement.

Thank you. We have added a section in the supplement titled "Adduct Ions" to address this.

In supplementary information:

- Mention in Figure S2f caption that the labels in the x axis are PubChem IDs.

Thank you for catching that. The caption has been updated.

- I am not sure how useful Figure S3 really is at these sizes.

Our goal with this figure is to convey the scale of the synthesis, even if structural details are not well resolved on a printed page. Readers viewing it as a PDF would have the opportunity to zoom in to see the full product structures.

- Figure S5c: it would be interesting to know why the left side of the curve is different from the rest.

The left-side looks different (less smooth), because the x-axis is on a cumulative log scale and as such there are fewer points on the left hand side, making the variation in delays between the first several transfers more apparent than the later ones (even though they are statistically similar).

- Figure S7: It would be helpful to label the different points in the plot and table with their reference/citation numbers.

Good point. We have added reference IDs (figure specific) so that plot points can be matched to their respective table entries.

- Supplementary section 3.4 would benefit from further discussion about how far technology can scale to smaller spots/less material. Or perhaps a more comprehensive discussion could follow Supplementary section 4.

We have added a mention of scaling limits to section 3.4 of the supplement (under 'Repeated Reads').

Reviewers' Comments:

Reviewer #1:

Remarks to the Author:

The authors have now thoroughly revised their manuscript. All the remarks of the reviewers have been carefully addressed. Important missing information has been included in the main text, which is now suitable for publication.

Reviewer #2:

Remarks to the Author:

Comments for Author:

The authors have addressed a large number of concerns raised. Here are the remaining concerns:

From authors: "We appreciate this comment, but we feel showing the masses of all 1,500 compounds would be cumbersome to read."

- Showing masses of all 1,500 compounds would be cumbersome to read if provided as a table in the text, but it would be well suited for a Supplementary electronic spreadsheet.

- The comparison with Flash is still misleading. While it may be true that building a single cell with 2^{100} states is impractical, building a 3D stack with 100 single-bit cells is possible and has been done. This is not something one could do with multicomponent molecular memories at practical access times.

- Having pointed this out, though, the observation that a single cell of multicomponent molecular memory can store a large number of bits is valid and should be highlighted, just not compared with flash itself because it implies a density argument that is still not fully formed. Perhaps the solution is to compare a multicomponent molecular memory cell to a generic cell of electronic or magnetic storage, instead of picking a particular technology. If a particular technology is chosen, the comparison should be more thorough.

- It would be ok to compare with flash or any other specific technology directly if the authors had hard evidence that multicomponent molecular memories scale to smaller feature sizes, but as the authors point out in the response letter, this is not something they can do just yet.

- Size of spots: I would encourage the authors to include the spot height in the caption of Figure 1, even if already mentioned in the Supplementary materials.

- Simulated error rates: it is still not clear how the 100 mixtures were corrupted at various bit error rates. Is this a digital corruption (flipping present/absent from the mix)? Is it balanced (i.e., as many present  absent as absent  present)? Does it reflect the actual corruption statistics seen in real systems?

From authors: "Even with raw error rates several times larger than the guaranteed threshold, the vast majority of errors could still be corrected."

- Please state how these errors are corrected. Is it just by looking at the distance to valid codewords, or is there any additional error correction being applied on top of this?

From authors: "The overall time to write a data plate ranged from 0.3 to 7.9 hours"

- It is useful to also use a metric of throughput, for example, how many bits can be written/read per minute. It would be interesting to contrast the different encodings and whether they have a significant impact on the write/read bit rate.

From authors: "Ugi 1439 is better identified with more complex ions containing unreacted starting materials"

- Isn't this concerning? If unreacted starting materials are being used to identify certain Ugis, then would these Ugis be distinguishable from others that are different combinations of the same subcomponents?

Dear Reviewers,

Thank you again for your supportive and constructive comments. Please find our responses below. Enclosed are revised versions of the manuscript and supplement with the new changes highlighted in blue.

Sincerely,
Chris Arcadia and Jacob Rosenstein

Reviewer #1 (Remarks to the Author):

The authors have now thoroughly revised their manuscript. All the remarks of the reviewers have been carefully addressed. Important missing information has been included in the main text, which is now suitable for publication.

Reviewer #2 (Remarks to the Author):

The authors have addressed a large number of concerns raised. Here are the remaining concerns:

From authors: "We appreciate this comment, but we feel showing the masses of all 1,500 compounds would be cumbersome to read."

- Showing masses of all 1,500 compounds would be cumbersome to read if provided as a table in the text, but it would be well suited for a Supplementary electronic spreadsheet.

Thank you. We have added a supplementary CSV file ("Ugi_Product_Library.csv") listing the components, masses, sodiated peak intensity, and reagents for each Ugi product.

- The comparison with Flash is still misleading. While it may be true that building a single cell with 2^{100} states is impractical, building a 3D stack with 100 single-bit cells is possible and has been done. This is not something one could do with multicomponent molecular memories at practical access times. Having pointed this out, though, the observation that a single cell of multicomponent molecular memory can store a large number of bits is valid and should be highlighted, just not compared with flash itself because it implies a density argument that is still not fully formed. Perhaps the solution is to compare a multicomponent molecular memory cell to a generic cell of electronic or magnetic storage, instead of picking a particular technology. If a particular technology is chosen, the comparison should be more thorough. It would be ok to compare with flash or any other specific technology directly if the authors had hard evidence that multicomponent molecular memories scale to smaller feature sizes, but as the authors point out in the response letter, this is not something they can do just yet.

We understand your point, and we have revised this paragraph. We have tried to clarify that the conceptual comparison is bits per cell, not overall spatial density, and that our molecular memory spots are currently much larger than electronic memory cells. We have also added a reference to multilayer flash memory.

- Size of spots: I would encourage the authors to include the spot height in the caption of Figure 1, even if already mentioned in the Supplementary materials.

We have described the dried spot height in the caption of Figure 1.

- Simulated error rates: it is still not clear how the 100 mixtures were corrupted at various bit error rates. Is this a digital corruption (flipping present/absent from the mix)? Is it balanced (i.e., as many present  absent as absent  present)? Does it reflect the actual corruption statistics seen in real systems?

For these simulations, chemical presence bits were flipped randomly across all mixtures to simulate different raw error rates. The corruption process was balanced, which reflects the fact that the experimental FPR and FNR in direct writes are similar (see Figure 4.c top). We have clarified the description in the text.

From authors: "Even with raw error rates several times larger than the guaranteed threshold, the vast majority of errors could still be corrected."

- Please state how these errors are corrected. Is it just by looking at the distance to valid codewords, or is there any additional error correction being applied on top of this?

Correct! The measured codeword is assigned the valid codeword nearest to it (Hamming distance). No additional error correction is used on top of this. We have highlighted this in a preceding sentence.

From authors: "The overall time to write a data plate ranged from 0.3 to 7.9 hours"

- It is useful to also use a metric of throughput, for example, how many bits can be written/read per minute. It would be interesting to contrast the different encodings and whether they have a significant impact on the write/read bit rate.

For a fixed library size, the fastest write times will come from the sparsest mappings, since the write speed bottleneck is in fluid handling and sparsity allows one present compound to represent more than one bit of data. The read time depends on several mass spectrometer settings. During the course of this work we made several adjustments to progressively improve the write/read processes (eg. sorting fluid transfer lists by source well), which means the precise write/read bit rates do not necessarily capture broader trends among the coding schemes. All of the descriptions for each plate are listed in Figure S7.

From authors: "Ugi 1439 is better identified with more complex ions containing unreacted starting materials"

- Isn't this concerning? If unreacted starting materials are being used to identify certain Ugis, then would these Ugis be distinguishable from others that are different combinations of the same subcomponents?

We have updated the text to clarify this item. It is true that different Ugis could theoretically produce complexes with other molecules, producing two peaks whose masses coincide. Yet even if this were to occur, these would be just two out of many candidate features, and the detection algorithms would learn to discount these peaks during training. The strength of our statistical readout approach is that it does not rely on just one mass feature. If we continue to increase the library size and mixture size, then at some point the spectra will be too crowded to reliably decode, but in our datasets so far we have not experienced this.

Reviewers' Comments:

Reviewer #2:

Remarks to the Author:

Thank you for carefully addressing all the points raised. This paper is ready for publication.